# Vision Learners Meet Web Image-Text Pairs

**Bingchen Zhao**[1], **Quan Cui**[2], **Hao Wu**[2], **Osamu Yoshie**[3], **Cheng Yang**[2], **Oisin Mac Aodha**[1]
[1] *University of Edinburgh*    [2] *Bytedance*    [3] *Waseda University*

**Reviewed on OpenReview:** *https://openreview.net/forum?id=OCnTRfmaTb*

## Abstract

Many self-supervised learning methods are pre-trained on the well-curated ImageNet-1k dataset. In this work, given the excellent scalability of web data, we consider self-supervised pre-training on noisy web sourced image-text paired data. First, we conduct a benchmark study of representative self-supervised pre-training methods on large-scale web data in a like-for-like setting. We compare a range of methods, including single-modal ones that use masked training objectives and multi-modal ones that use image-text constrastive training. We observe that existing multi-modal methods do not outperform their single-modal counterparts on vision transfer learning tasks. We derive an information-theoretical view to explain these benchmark results, which provides insight into how to design a novel vision learner. Inspired by this insight, we present a new visual representation pre-training method, MUlti-modal Generator (MUG), that learns from scalable web sourced image-text data. MUG achieves state-of-the-art transfer performance on a variety of tasks and demonstrates promising scaling properties.

## 1 Introduction

Self-supervised representation learning (SSL) has attracted a considerable amount of attention recently as it offers the potential to reduce the reliance on labor-intensive human collected annotations when training models. The design of SSL methods is mainly guided by the InfoMax principle (Hjelm et al., 2019), which shows that maximizing the mutual information $I(X; Z)$ of the input data $X$ and the learned representation $Z$ can lead to better representations. In the case of visual data, discriminative contrastive learning (He et al., 2020; Chen et al., 2020d; 2021; 2020b;c; Shen et al., 2022; Bardes et al., 2022) and generative masked image modeling (Bao et al., 2022; He et al., 2022; Baevski et al., 2022; Wei et al., 2022; Peng et al., 2022) have been demonstrated to learn transferable representations from images by attempting to solve pre-defined pretext objectives that aim to indirectly optimize $I(X; Z)$, achieving state-of-the-art results on popular computer vision benchmarks. Despite these remarkable advancements, most SSL methods are developed on the ImageNet-1k (Deng et al., 2009) dataset, which is well-curated and thus not representative of many real world use cases. As a result, these works do not fully exploit the potential scalability of SSL to larger uncurated datasets, *e.g.*, many existing methods are only evaluated on the ImageNet-1k dataset (Xie et al., 2021a; He et al., 2020; 2022). There exists works on extending SSL to larger uncurated datasets (Wen et al., 2022; Xie et al., 2021a; Hénaff et al., 2021; Xie et al., 2021b; Wang et al., 2021), yet most of these works focus on how to obtain object centric representations (Wen et al., 2022), or target one specific downstream task, such as object detection (Wang et al., 2021; Xie et al., 2021b;a; Hénaff et al., 2021).

In contrast to the previous single-modal SSL methods, pioneering multi-modal SSL methods (*e.g.*, CLIP (Radford et al., 2021) and ALIGN (Jia et al., 2021)) have shown impressive scalability due to their ability to train on easily collected large-scale web sourced image-text pairs (Sharma et al., 2018; Thomee et al., 2016; Schuhmann et al., 2021). Multiple works have shown that the textual information present in web image-text pairs can greatly benefit various downstream tasks (Mu et al., 2022; Kim et al., 2021; Yao et al., 2022; Li et al., 2022b) and thus contribute to improved transfer learning performance. However, many of these existing methods differ not only in the training losses they use, but also in the terms of the model architectures and training data, which makes direct comparison difficult. To address this, in this work we

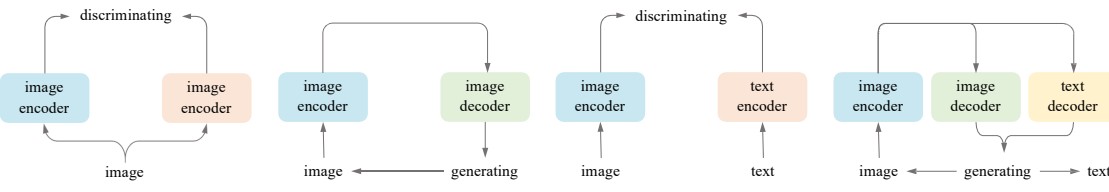

Figure 1: Comparison of different vision pre-training paradigms that use images or image-text pairs. Four paradigms are considered: (i) single-modal discriminative (*e.g.*, SimCLR (Chen et al., 2020b)), (ii) single-modal generative (*e.g.*, MAE (He et al., 2022)), (iii) multi-modal discriminative (*e.g.*, CLIP (Radford et al., 2021)), and (iv) our proposed multi-modal generative approach named **MUG**. The multi-modal generative paradigm simultaneously generates images and text using only image representations.

conduct a benchmark study of single-modal SSL methods (*i.e.*, with only images) and multi-modal SSL methods (*i.e.*, with image-text pairs) using the same web-sourced image-text datasets and training settings.

Based on our benchmark study, we make two key observations. First, generative methods achieve the best transfer performance. Second, and more surprisingly, the representations learned from multi-modal SSL methods do not outperform single-modal SSL ones. To explain these observations, we explore an information-theoretical motivated perspective. Starting from the InfoMax principle (Hjelm et al., 2019) for learning representations, we use the information-bottleneck theory to explain the transferability of deep representations (Tishby & Zaslavsky, 2015; Shwartz-Ziv & Tishby, 2017; Alemi et al., 2017; Cui et al., 2022a). We show that generative methods approach a better upper bound on the mutual information $I(X; Z)$, and that introducing multiple generative objectives can further improve upon this bound.

In Fig. 1, we group previous representative methods into three families: (i) single-modal discriminative (*e.g.*, SimCLR (Chen et al., 2020b)), (ii) single-modal generative (*e.g.*, MAE (He et al., 2022)), and (iii) multi-modal discriminative (*e.g.*, CLIP (Radford et al., 2021)). Empirical observations and theoretical analyses jointly inspire us to design a fourth paradigm, multi-modal generative pre-training (see Fig. 1 (iv)). Under this paradigm, we propose a novel vision learner for web image-text data pre-training, named **MU**lti-modal **G**enerator (MUG). MUG is composed of purely multi-modal generative objectives that attempt to generate the multi-modal joint distribution from a single-modal image input. From an information-theoretical view, we show that MUG has a better upper bound for feature transferability, which helps the vision encoder learn more transferable representations when compared to alternative paradigms.

We present empirical results across a variety of downstream vision tasks to further validate the advantages of MUG. For example, MUG outperforms the previous best performing methods by 2.0% mIoU when transferred to the ADE20k benchmark and by 0.5% on ImageNet-1k classification. Extensive ablations are conducted to help better illuminate the impact of critical model and design choices. Finally, visualizations are provided that illustrate that our learned representations are capable of capturing the joint image-text distribution.

## 2 Related Work

There is a large body of existing work on the topic of self-supervised learning of representations from visual data. We focus on constrastive and masked-based learning objectives, and also discuss learning from paired image and text data.

### 2.1 Vision Learners with Contrastive Learning

Contrastive learning is a commonly used paradigm for single-modal self-supervised representation learning. Here, the pretext task is an instance discrimination one (Wu et al., 2018) which learns a representation by pulling positive examples that are generated from augmentations of the same image closer, while pushing negative examples (*e.g.*, different images from the same mini-batch) apart in the embedding space. Building on the basic instance discrimination task, SimCLR (Chen et al., 2020b;c) showed that larger mini-batches and stronger augmentations can result in more effective representations. MoCo (He et al., 2020; Chen et al.,

2020d; 2021) showed that introducing more negative examples using a momentum encoder and memory bank results in representations that are even more effective for downstream transfer. A variety of techniques for improving contrastive learning have been proposed, including hard example mining (Kalantidis et al., 2020; Zhao & Wen, 2020), image mixing (Shen et al., 2022; Zhu et al., 2021), and by injecting localized priors (Hénaff et al., 2021; Wen et al., 2022; Xie et al., 2021b; Wang et al., 2021).

The underlying principle of contrastive learning is the InfoMax principle (Hjelm et al., 2019), which posits that the mutual information $I(X; Z)$ between the input image $X$ and the encoded representation $Z$ should be maximized so that the representation of an image can uniquely identify it from the entire dataset. CTC (Cui et al., 2022a) showed that the mutual information $I(X; Z)$ is the key to the transferability of one representation, and that preventing the decrease of $I(X; Z)$ during training can improve the transferability of the learned representation. CTC (Cui et al., 2022a) also showed that contrastive learning objectives are in practice learning to optimize $I(X; Z) = H(Z) - H(X|Z)$ by increasing $H(Z)$.

$$\underbrace{I(X; Z)}_{\text{Representation Learning}} = \underbrace{H(Z) - H(X|Z)}_{\text{Contrastive Learning}} = \underbrace{H(X) - H(X|Z)}_{\text{Generative Learning}} \tag{1}$$

## 2.2 Vision Learners with Masked Image Modeling

Inspired by the success of masked language modeling (MLM) using transformers (Vaswani et al., 2017) for natural language processing (Devlin et al., 2018; Brown et al., 2020), masked *image* modeling (Chen et al., 2020a; Bao et al., 2022; He et al., 2022; Xie et al., 2022) has been explored in the context of learning representations using vision transformers (Dosovitskiy et al., 2021; Touvron et al., 2020). Specifically, iGPT (Chen et al., 2020a) learns a representation by predicting unknown pixels from a sequence of pixels. Instead of operating in pixel space, BEiT (Bao et al., 2022) predicts a set of discrete tokens from the encoder of DALLE (Ramesh et al., 2022; Van Den Oord et al., 2017), where the tokens are encoded versions of image patches as in (Ramesh et al., 2021). The training task was further simplified by MAE (He et al., 2022) and SimMIM (Xie et al., 2022), which only enforce the model to reconstruct the masked pixels of the input image using a $\ell_2$ loss and do not require the use of discrete token encoders.

The pretext task of masked image modeling can be viewed as optimizing $-\log p(X|Z)$, where $Z$ is the representation of the non-masked image $X$ and $p(\hat{X}|Z)$ is the likelihood of generating the masked image parts $\hat{X}$ given the learned representation of the input tokens. This is equivalent to minimizing $H(X|Z)$. Given that $I(X; Z) = H(X) - H(X|Z)$, we can see that masked image modeling still follows the InfoMax (Hjelm et al., 2019) principle, where the maximization of $I(X; Z)$ is performed by optimizing $H(X|Z)$, instead of $H(Z)$ as in contrastive learning. MIM has been shown to be more effective at learning transferable representations compared to contrastive learning (He et al., 2022; Xie et al., 2022), indicating the effectiveness of generative pre-training.

## 2.3 Vision Learners with Web Image-Text Pairs

The vast majority of the previously discussed works have explored SSL in the context of single-modal well-curated image datasets such as ImageNet-1k (Deng et al., 2009). However, it is relatively easy to collect large-scale datasets containing image-text paired data from the web, *e.g.*, images and associated text captions (Sharma et al., 2018; Schuhmann et al., 2021). Many approaches have recently began to study what impact this type of data has on learning visual representations, *e.g.*, Radford et al. (2021); Mu et al. (2022); Kim et al. (2021); Yao et al. (2022); Li et al. (2022b); Yu et al. (2022); Desai & Johnson (2021). CLIP (Radford et al., 2021) explores this challenge by performing contrastive learning over paired image and text data. By leveraging separate visual and text encoders, their training objective aims to maximize the similarity between the learned embeddings of the paired image-text data from each training instance and minimize the similarity between unpaired images and text. SLIP (Mu et al., 2022) builds on this, by learning the representation by jointly performing contrastive learning on both image-text paired datasets like CC3M (Sharma et al., 2018) and image-only datasets like ImageNet (Deng et al., 2009) using strong augmentations (Chen et al., 2020b). SLIP showed that the added self-supervised signal on image-only

datasets can improve the performance for zero-shot image classification and transfer learning. However, as we will show later in our like-for-like experiments, the benefit of SLIP over the baseline CLIP model is limited, we conjecture this is due to the fact that the two objectives of SLIP are both contrastive learning losses which are discriminative. In Table 1, we show that discriminative objectives are not as effective as generative ones.

CoCa (Yu et al., 2022) poses representation learning as a cross-modal contrastive learning and auto-regressive caption generation task. The contrastive loss of CoCa contrast the image embedding with its corresponding textual caption embedding. The caption generation loss is designed to first encode the text only features using a text encoder with causal attention, and then a transformer decoder with cross-attention to image encoded features are used to decode the caption auto-regressively. These two objectives both enforce the model to learn a predictive image representation of the caption. CapPa (Tschannen et al., 2023) further explores visual representation learning using only an auto-regressive image captioning loss. High performance of zero-shot learning and few-shot linear classification is achieve in CapPa. In contrast, in our designed model in Section 4, we enforce the image representation to not only be predictive of the caption, but also to be predictive of the image itself. Additionally, this paper also provides an intuitive understanding of why and how such generative training objective results in a strong representation learning ability from an information theoretical perspective.

There also exists other work that focuses on achieving higher zero-shot performance from a frozen visual representation (Zhai et al., 2022) by filtering the caption to achieve better performance (Li et al., 2022a), or adapting frozen visual and language models together to achieve in-context learning abilities on complex visual-language tasks (Alayrac et al., 2022; Li et al., 2023; Koh et al., 2023; Zong et al., 2024). In this work, we first comprehensively study existing paradigms for learning transferable representations from publicly available image-text paired datasets. Inspired by this, we propose a new method for learning representations and further compare our method to the previous state-of-the-art works where we out perform them across a wide variety of tasks.

## 3 Preliminaries

### 3.1 Benchmarking SSL Methods on Web Data

Given a dataset of image-text pairs $\mathcal{D} = \{(x_i^V, x_i^L)\}$ sampled from the joint distribution of $p(X^V, X^L)$, where an image is of size $x_i^V \in \mathbb{R}^{C \times H \times W}$, and a piece of text is of size $x_i^L \in \mathbb{R}^L$, our goal is to learn a visual feature encoder $f : \mathbb{R}^{C \times H \times W} \to \mathbb{R}^D$ that maps an input image into a compact feature embedding and enables effective transfer learning performance (He et al., 2019).

Two families of self-supervised learning methods can be used to achieve this goal. This first is single-modal SSL, which uses only the images $x_i^V$ within the dataset $\mathcal{D}$. Section 2.1 and Section 2.2 introduce and describe the current most representative methods for single-modal SSL. The second approach is multi-modal SSL. This aims to learn an effective representation by aligning the representation of the input image with the corresponding paired input text. A brief introduction to multi-modal SSL methods is provided in Section 2.3. As a preliminary motivation for our work, we first perform a benchmark analysis. We pre-train representative SSL methods on the *same* web image-text dataset, for the *same* number of training epochs. Then, we evaluate the transfer performance on downstream tasks by fine-tuning the resulting pre-trained models.

**Experimental setup.** We categorize methods according to their pre-training paradigm, as discussed in Fig. 1. For single-modal SSL methods, we choose MoCoV3 (Chen et al., 2021) and SimCLR (Chen et al., 2020b) as representative discriminative methods, and MAE (He et al., 2022) as a representative generative method. For image-text multi-modal SSL, we choose CLIP (Radford et al., 2021) and SLIP (Mu et al., 2022) as discriminative methods, and also include CoCa (Yu et al., 2022), which is composed of both discriminative and generative targets. Notably, in addition to these methods, we further implement another multi-modal baseline namely Masked Captioner (MAC), which generates text from masked images. It is trained using only a purely generative cross-modal objective. The details of MAC will be introduced in Section 4. For the pre-training hyper-parameters, we strictly follow the recommended setting in the original paper of each method. For a fair comparison, we pre-train all methods on a web dataset CC3M (Sharma

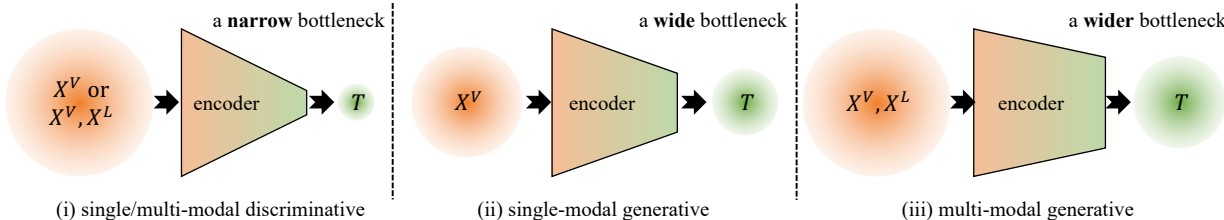

Figure 2: **Left:** Single/multi-modal discriminative methods have a narrow bottleneck and thus learn a less informative representation. **Middle:** Single-modal generative methods have a wide bottleneck and thus learn a more informative representation. **Right:** Multi-modal generative methods have a wider bottleneck for generating (*e.g.*, recovering) the joint distribution of both modalities, and as a result learn an even more informative representation.

Table 1: Like-for-like comparison of existing SSL pre-training methods. All models are pre-trained on the CC3M dataset (Sharma et al., 2018), and evaluated on ImageNet-1k (IN-1K) (Deng et al., 2009) via end-to-end fine-tuning. "gen. and dist." are short hand for generative and discriminative. "$\mathcal{I}$ and $\mathcal{T}$" denote images and text. For instance, SimCLR (Chen et al., 2020b) is pre-trained by discriminating images (disc. $\mathcal{I}$). CLIP (Radford et al., 2021) is pre-trained by discriminating image-text pairs (disc. $\mathcal{I}\&\mathcal{T}$). CoCa (Yu et al., 2022) has two objectives: discriminating image-text pairs (disc. $\mathcal{I}\&\mathcal{T}$) and generating text from images (gen. $\mathcal{T}$ w. $\mathcal{I}$). We also include a MAsked Captioner (MAC) baseline that generates text from masked image patches (gen. $\mathcal{T}$ w. $\mathcal{I}$).

| method | pre-training paradigm | IN-1K (Deng et al., 2009) |
|---|---|---|
| *single-modal pre-training* | | |
| MAE (He et al., 2022) | **gen. $\mathcal{I}$** | **83.0** |
| SimCLR (Chen et al., 2020b) | disc. $\mathcal{I}$ | 82.7 |
| MoCoV3 (Chen et al., 2021) | disc. $\mathcal{I}$ | 82.6 |
| *multi-modal pre-training* | | |
| MAC | **gen. $\mathcal{T}$ w. $\mathcal{I}$** | **81.7** |
| CLIP (Radford et al., 2021) | disc. $\mathcal{I}\&\mathcal{T}$ | 79.7 |
| SLIP (Mu et al., 2022) | disc. $\mathcal{I}\&\mathcal{T}$ + disc. $\mathcal{I}$ | 80.9 |
| CoCa (Yu et al., 2022) | disc. $\mathcal{I}\&\mathcal{T}$ + gen. $\mathcal{T}$ w. $\mathcal{I}$ | 79.5 |

et al., 2018) for 400 epochs. When transferring the representation to ImageNet-1k (Deng et al., 2009), we follow the widely used fine-tuning recipe introduced by Bao et al. (2022); He et al. (2022).

## 3.2 Key Observations

We make the following key observations from the results of the experiments in Table 1:

**Generative methods (*e.g.*, MAE) achieve the best results.** Across all methods, we can observe that MAE (He et al., 2022) achieves the best performance. As mentioned in Section 2.1 and Section 2.2, we argue that discriminative pre-training and generative pre-training approach the InfoMax (Hjelm et al., 2019) principle via different mechanisms. Specifically, discriminative pre-training optimizes $I(X; Z_d) = H(Z_d) - H(Z_d|X)$ via maximizing $H(Z_d)$, while generative pre-training optimizes $I(X; Z_g) = H(X) - H(X|Z_g)$ via minimizing $H(X|Z_g)$ (Cui et al., 2022a), here $Z_d$ is the representation learned by discriminative pre-training, and $Z_g$ is the representation learned by generative pre-training. Suppose the models learned by generative pre-training and discriminative pre-training both achieve zero training loss, then $I(X; Z_d)$ will be $H(Z_d)$ and $I(X; Z_g)$ will be $H(X)$. In the discriminative pre-training scenario, if we consider the Markov Chain of $X \rightarrow Z_d$, we have $I(X; X) \geq I(X; Z_d)$ from the data process inequality, *i.e.*, $H(X) \geq I(X; Z_d) = H(Z_d)$. Thus generative pre-training has a better transferability upper bound compared to discriminative pre-training, and results in more informative and beneficial features for downstream tasks.

$$\text{Disc.: } I(X; Z_d) = \underbrace{H(Z_d)}_{\text{maximizing}} - H(Z_d|X) \xrightarrow{\text{upperbound}} H(Z_d) \tag{2}$$

$$\text{Gen.: } I(X; Z_g) = H(X) - \underbrace{H(X|Z_g)}_{\text{minimizing}} \xrightarrow{\text{zero training loss}} H(X) \tag{3}$$

**Current multi-modal methods do not outperform single-modal ones.** We can observe that the best single-modal pre-training method outperforms the best multi-modal pre-training method by 1.3% on the ImageNet-1k accuracy. Compared with our introduced MAC baseline, multi-modal discriminative methods (*e.g.*, CLIP and CoCa) yield worse results. We believe the reason is that *the information from the text modality is greatly limited.* A typical text annotation only contains partial information about the contents of the corresponding image. A prominent researcher has commented that "language is a very low-bandwidth method for transmitting information: isolated words or sentences, shorn of context, convey little." (Browning & LeCun, 2022) It suggests the $H(X^L)$ could be much lower than $H(X^V)$. In the case of the multi-modal discriminative method CLIP, a low $H(X^L)$ directly limits $I(X^L; Z_d)$. Thus, poor transferring performance is observed. When comparing SimCLR and SLIP we observe that SLIP falls behind SimCLR by 1.8% accuracy, which further supports the potential limitations of multi-modal discriminative training objectives. However, MAC is generative and alleviates this problem, but it still cannot outperform single-modal methods.

$$H(X^L) \ll H(X^V) \tag{4}$$

**Discussions and opportunities.** Inspired by the information-bottleneck theory (Tishby & Zaslavsky, 2015; Shwartz-Ziv & Tishby, 2017; Alemi et al., 2017; Cui et al., 2022a; Hjelm et al., 2019), we illustrate to help understand our observations in Fig. 2. The first observation is that generative methods can help the model learn more transferable visual representations compared to discriminative ones due to a superior $I(X; Z)$ upper bound. In Fig. 2 (left), we argue that discriminative methods have a narrow bottleneck and a low $I(X; Z)$, thus a less informative representation learned. More detailed discussions are provided in Appendix Appendix A.6.

In Fig. 2 (middle), generative methods have a wider bottleneck, leading to more informative representations. Given the side effects of multi-modal discriminative objectives (the second observation), developing multiple generative objectives will encourage the model to learn more informative representations. As shown in Fig. 2 (right), for an image-text pair, generating the joint distribution of both the image and text modality will help the model to "absorb" as much information as possible from the data and approach a larger $I(X; Z)$.

Note that here we are using the term bottleneck to refer to the variance in the representation space $Z$ with respect to the input images $X$, *i.e.*, how much the representation changes when the input is changed. One way to understand this bottleneck is from the perspective of the rank of the feature matrix (Garrido et al., 2022). A higher rank would indicates that the model can capture more variance in the input, whereas a collapsed lower rank captures less information about the input. In contrast, a completely collapsed representation would map all inputs to a same feature which has a rank of zero. In Appendix A.4, we provide a qualitative measure of the rank of different pre-trained models.

## 4 Approach

### 4.1 Motivation

Following the above observations, in this section we describe a more effective vision learner that learns from web sourced image-text paired training data. We first formulate the mechanism that a multi-modal generative objective uses to learn a more transferable representation, in line with Fig. 2 (right). We believe that introducing multiple generative objectives is feasible, where now the optimization objective is to learn

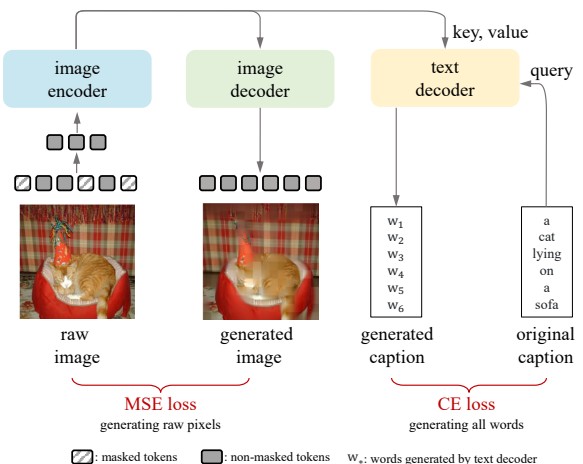

Figure 3: Illustration of our multi-modal self-supervised approach MUG.

**Algorithm 1** Pseudocode for MUG.

```
# img, txt: image-text paired data
# txt_mask: causal mask for captioning
def text_decoder(q, kv, mask):
    q = single_modal_attn(q, mask)
    cap_res = multimodal_attn(q, kv)
    return cap_res

patch_img = patchify(img)
masked_token = masking(patch_img) # [N, L, D]
latent = vit_encoder(masked_token) # [N, L, D]

# Generative objective for image
recon_img = mae_decoder(latent) # [N, L, D]
recon_loss = mse_loss(img, recon_img)

# Generative objective for text
label, txt = txt[:, 1:, :], txt[:, :-1, :]
txt_feat = tokenizer(txt) # [N, L, D]
cap_res = text_decoder(q=txt_feat, kv=latent, mask=
    txt_mask)
cap_loss = ce_loss(label, cap_res)

loss = recon_weight * recon_loss + cap_weight *
    cap_loss
```

how to generate the joint distribution $p(X^V, X^L)$ from a single masked input distribution $\hat{X}^V$ as our focus is on learning a visual encoder.

To generate the joint distribution $p(X^V, X^L)$ from the latent representation $Z = f(\hat{X}^V)$ extracted from the non-masked visual input $\hat{X}^V$, the model maximizes the following mutual information:

$$
\begin{aligned}
I(X^V, X^L; Z) =& H(X^V, X^L) - H(X^V, X^L|Z) \\
\geq& \max\left[H(X^V), H(X^L)\right] - H(X^V, X^L|Z) \quad (5) \\
\rightarrow& \max\left[H(X^V), H(X^L)\right] \text{ (zero training loss)} \quad (6)
\end{aligned}
$$

If we consider the fully optimized model to the optimal, then $H(X^V, X^L|Z) = 0$. As the lower bound of $I(X^V, X^L; Z)$ is greater than or equal to the lower bound of $I(X^V; Z)$ because of the added variable $X^L$, we can observe that generating the joint distribution of $p(X^V, X^L)$ can at least learn an equally transferable representation as single-modal generative pre-training. In practice, we observe that multi-modal generative pre-training achieves superior transferability. This design motivation is also in line with the discussion in Section 3.2. Overall this intuition motivates us to design a generative pre-training framework with multiple generation tasks using only the visual input and representation.

### 4.2 MUG: MUlti-modal Generator

Here we describe our novel multi-modal generative vision learner, named MUlti-modal Generator (MUG). The framework of MUG is presented in Fig. 3. It consists of an image encoder, an image decoder, and a text decoder. Two tasks are involved, *i.e.*, raw pixel and original caption generation.

**Raw pixel generation.** Given a masked visual input $\hat{x}_i^V = M \odot x_i^V$, where $M$ is the random generated 0-1 mask following the procedure in MAE (He et al., 2022) where 0 indicates that the image region is masked out and 1 indicates that it is visible to the model, $\odot$ denotes the element-wise product, the visual feature encoder $f$ maps the masked input image to a latent representation $t_i^V = f(\hat{x}_i^V)$. The latent representation $t_i^V$ is then fed into a visual decoder $g_V$ to generate $\tilde{x}_i^V = g_V(t_i^V)$ which is the reconstruction of the non-masked visual input. We adopt the reconstruction loss from MAE (He et al., 2022) to train both $f$ and $g_V$:

$$
\mathcal{L}_V = \frac{1}{\Omega(\hat{x}_i^V)} \|(1 - M) \odot x_i^V - (1 - M) \odot \tilde{x}_i^V\|^2, \quad (7)
$$

where $\Omega(\hat{x}_i^V)$ is the number of masked elements in $\hat{x}_i^V$. This loss encourages the generation of the visual part of the input.

**Caption generation.** We further propose to generate the paired text (*i.e.*, the image caption) $x_i^L$ from the feature $t_i^V$ of the masked visual input $\hat{x}_i^V$. To do this, we feed the feature $t_i^V$ to a text decoder $g_L$ to generate the text captions $\tilde{x}_{i,j}^L = g_L(x_{i,<j}^L, t_i^V)$ in an auto-regressive fashion, driven by the following loss:

$$\mathcal{L}_L = -\sum_{j=1}^J \log P(\tilde{x}_{i,j}^L | x_{i,<j}^L, t_i^V), \tag{8}$$

where $J$ is the maximum length of a textual caption, $x_{i,<j}^L$ is the first $j-1$ tokens of the caption and $x_{i,j}^L$ is the $j$-th token of the caption. The text generation is conditioned on the image features, which helps to improve the transferability of the image encoder. Thus, the proposed text decoder has two parts, *i.e.*, single-modal and multi-modal layers. The single-modal part is constructed as an auto-regressive model via a causal mask (Vaswani et al., 2017), where each word can only attend to the words before itself in a caption sequence. It only provides the text decoder with contextual information when generating the caption. We design the multi-modal part using the image features as the *key* and *value* and the output from single-modal layers as the *query* in a cross-attention layer (Vaswani et al., 2017). This ensures the text generation is derived from the image features as the contextual information only serves as the *query*. Thus, the entire text decoder can be seen as regrouping the image features $t_i^V$ with contextual information to generate captions.

**Optimization.** Our multi-modal pre-training framework is trained by optimizing the above two losses:

$$\mathcal{L} = \lambda_V \mathcal{L}_V + \lambda_L \mathcal{L}_L, \tag{9}$$

where $\lambda_V$ and $\lambda_L$ are different weights for the image and text generation losses. Additionally, for the MAC method we evaluate in Section 3, only the caption loss $\mathcal{L}_L$ is used to optimize the model.

### 4.3 Understanding the Loss Function

If we consider the MUG image decoder as modeling the image using a Gaussian prior (Zhao et al., 2015), then the loss in Eq. (7) is actually maximizing the log-likelihood of the generated image:

$$\log P_{x_i^V}(\tilde{x}_i^V) = \log \frac{1}{\sigma\sqrt{2\pi}} \exp\left(-\frac{1}{2}\left(\frac{\tilde{x}_i^V - x_i^V}{\sigma}\right)^2\right) = -\frac{1}{\text{M}}\|\tilde{x}_i^V - x_i^V\|^2 - \text{C}, \tag{10}$$

where $\sigma$ is the standard deviation of the distribution. M and C are constants that do not affect the optimization process. Together with $\mathcal{L}_L$, we can see that our loss function in Eq. (9) is maximizing the log-likelihood of both the generated image and the generated paired text. Thus the latent representation $Z$ extracted by the model is optimized to be more predictive of the image $X^V$ and the caption $X^L$. In other words, the conditional entropy $H(X^V, X^L|Z)$ is minimized via the two losses in our framework, maximizing the mutual information $I(X^V, X^L; Z)$, and leading to a more transferable representation.

## 5 Experiments

### 5.1 Implementation Details

**Encoders and decoders.** Our framework contains an image encoder $f$, and two decoders $g_V$ and $g_L$ for decoding the latent representation to reconstruct the image and the textual caption, respectively. The image encoder $f$ is implemented using a Vision Transformer (Dosovitskiy et al., 2021) backbone, and the image decoder is implemented with the same architecture as in MAE (He et al., 2022). We implement the text decoder $g_L$ following SimVLM (Wang et al., 2022). For efficient training, we train our encoder-decoder architecture using teacher-forcing (Williams & Zipser, 1989). Detailed implementation details, including architectures and hyper parameters, are provided in the Appendix. The code is available at https://huggingface.co/spaces/tennant/MUG_caption.

**Pre-training datasets.** We train models on the publicly available CC3M (Sharma et al., 2018) and LAION400M (Schuhmann et al., 2021) datasets. Additionally, we also collect 200M web image-text pairs to study the scalability of MUG. The collected data has no filtering processing and as a result it potentially contains a large amount of noise which presents a challenge for multi-modal methods. We denote this privately collected dataset as W200M.

Table 2: Comparison of different methods pre-trained on CC3M (Sharma et al., 2018) and then evaluated via fine-tuning on ImageNet-1k (IN-1K) (Deng et al., 2009). Top-1 accuracy results are reported. In addition, we evaluate the IN-1k fine-tuned models on ImageNet-Adversarial (IN-A) (Hendrycks et al., 2021b) and ImageNet-Rendition (IN-R) (Hendrycks et al., 2021a) to assess out-of-distribution performance.

| method | data | #epoch | IN-1K | IN-A | IN-R |
|---|---|---|---|---|---|
| *single-modal pre-training* | | | | | |
| SimCLR | CC3M | 400 | 82.7 | 31.2 | 49.1 |
| MoCoV3 | CC3M | 400 | 82.6 | 29.7 | 46.9 |
| MAE | CC3M | 400 | 83.0 | 32.9 | 49.4 |
| *multi-modal pre-training* | | | | | |
| CLIP | CC3M | 400 | 79.7 | 23.6 | 42.6 |
| SLIP | CC3M | 400 | 80.5 | 20.5 | 44.4 |
| CoCa | CC3M | 400 | 79.5 | 18.1 | 42.9 |
| MAC (impl.) | CC3M | 400 | 81.7 | 27.5 | 44.3 |
| MUG (ours) | CC3M | 400 | **83.5** | **36.3** | **50.4** |

## 5.2 Transfer Learning

We begin by evaluating the transfer performance of different learned representations on a wide range of downstream tasks. Transfer learning is performed with conventional full end-to-end fine-tuning of the model weights on target task (He et al., 2022). This highlights the superiority of MUG compared to other methods.

**Image classification on ImageNet-1k.** First, we transfer learned representations to general image classification on ImageNet-1k (Deng et al., 2009) using end-to-end fine-tuning. In addition to reporting performance on the validation images in ImageNet-1k, we also evaluate ImageNet-1k fine-tuned models on out-of-distribution datasets ImageNet-Adversarial (IN-A) (Hendrycks et al., 2021b) and ImageNet-Rendition (IN-R) (Hendrycks et al., 2021a). Results are reported in Table 2 and Table 3. Compared to previous methods, MUG achieves the best top-1 accuracy 83.5% (with CC3M).When training MUG with ViT-S, ViT-B, and ViT-L backbones, we also observe consistent improvements over the previous single-modal best performing MAE (He et al., 2022). Both comparisons illustrates that MUG surpasses previous methods by making use of scalable web image-text paired data. Notably for the results on IN-A and IN-R, MUG outperforms the previous multi-modal methods by a large margin (*i.e.*, >10% on IN-A) even through our model leverages the same supervision signal as these other multi-modal methods. This indicates that MUG is much more effective for learning representations that generalize to hard out-of-distribution cases.

Table 3: Comparison of different backbone sizes (*i.e.*, ViT-S, ViT-B, and ViT-L) for models pre-trained with CC3M for 400 epochs and then fine-tuned on IN-1K. Top-1 accuracy is reported. Across all backbone sizes, we observe that MUG is superior to MAE in almost all cases.

| method | backbone | IN-1K | IN-A | IN-R |
|---|---|---|---|---|
| MAE | ViT-S | 80.9 | 22.7 | **44.1** |
| MUG (ours) | ViT-S | **81.6** | **25.0** | 42.5 |
| MAE | ViT-B | 83.0 | 32.9 | 49.4 |
| MUG (ours) | ViT-B | **83.5** | **36.3** | **50.4** |
| MAE | ViT-L | 85.0 | 49.4 | 59.6 |
| MUG (ours) | ViT-L | **85.4** | **53.8** | **63.1** |

Next, we evaluate pre-trained models using linear probing evaluation on ImageNet-1k. Results are reported in Table 4 and Table 6. Compared with discriminative pre-training methods, most generative methods obtain unsatisfactory IN-1k linear probing results. MUG also suffers from unsatisfactory linear probing performance common to other masked transformer models (He et al., 2022), where it has been hypothesized that linear probing requires discriminative representations, which cannot be satisfied by generative methods. For instance, SimCLR and MUG achieve 63.3% and 61.3% on the IN-1k validation set, respectively. However, in the case of IN-A and IN-R, generative methods like MUG achieve the best results. We conjecture this is due to the fact that discriminative methods can easily be overfit to the in-domain data, but conversely cannot achieve good out-of-distribution generalization. Additionally, compared with the original MAE, MUG outperforms it by a large margin. Notably, when using the backbone ViT-S in Table 6, our method outperforms MAE by **11.3%** top-1 accuracy. This indicates that MAE pre-trained models

can suffer from poor results due lower model capacity, whereas MUG pre-trained models can alleviate this drawback with the help of external data information from the text modality.

Table 4: Comparison of pre-training methods on ImageNet-1k (IN-1K) using linear probing. Top-1 accuracy is reported.

| method | data | #epoch | IN-1K | IN-A | IN-R |
|---|---|---|---|---|---|
| *single-modal pre-training* | | | | | |
| SimCLR | CC3M | 400 | **63.3** | 12.5 | 23.0 |
| MAE | CC3M | 400 | 57.4 | 9.1 | 17.8 |
| *multi-modal pre-training* | | | | | |
| CLIP | CC3M | 400 | 55.1 | 11.2 | 18.9 |
| CoCa | CC3M | 400 | 56.9 | 13.8 | 20.3 |
| MAC (impl.) | CC3M | 400 | 59.3 | 15.2 | 21.3 |
| MUG (ours) | CC3M | 400 | 61.3 | **16.7** | **24.2** |

Table 5: Comparison of pre-training methods when fine-tuning on fine-grained iNat-17, iNat-18, and P365. Top-1 accuracy is reported. MAE∗ is pre-trained on IN-1K.

| method | backbone | iNat-17 | iNat-18 | P365 |
|---|---|---|---|---|
| MAE | ViT-S | 65.4 | 70.0 | 57.9 |
| MUG (ours) | ViT-S | **65.5** | **70.7** | **58.3** |
| MAE∗ | ViT-B | 70.5 | 75.4 | 57.9 |
| MAE | ViT-B | 67.9 | 73.0 | 58.3 |
| MUG (ours) | ViT-B | **68.4** | **73.4** | **58.6** |
| MAE | ViT-L | 73.0 | 76.8 | 59.4 |
| MUG (ours) | ViT-L | **75.1** | **77.8** | **59.8** |

**Fine-grained image classification.** To illustrate that our pre-trained models do not over-fit on ImageNet-1k-based evaluations, we further transfer learned representations to fine-grained image datasets. Specifically, we report performance on iNaturalist-17 (iNat-17) (Van Horn et al., 2018), iNaturalist-18 (iNat-18), and Places365-Small (P365) (Zhou et al., 2017a) with end-to-end fine-tuning. Results are presented in Table 5. Similar to the results of ImageNet-1k, MUG still outperforms the previ-

Table 6: Comparison of different backbone sizes (*i.e.*, ViT-S, ViT-B, and ViT-L). Models are pre-trained on CC3M for 400 epochs and evaluated using linear probing on IN-1K.

| method | ViT-S | ViT-B | ViT-L |
|---|---|---|---|
| MAE | 41.0 | 57.4 | 63.5 |
| MUG (ours) | **52.3** | **61.3** | **67.6** |

ous SOTA method MAE on all evaluated cases. This further validates the success and robustness of MUG in transferring to image classification tasks. Moreover, we notice that the performance of MUG is relevant to the pre-training data domain. Concretely, iNat-17 and iNat-18 share a similar domain with IN-1K. Thus, the IN-1K MAE models perform better than CC3M MAE models. P365 shares a similar domain with CC3M, which leads to better results of CC3M-based models. Regardless of the fine-tuning data domain, the improvements from MUG are stable and generalize across datasets.

**Semantic segmentation on ADE20K.** We transfer MUG to a semantic segmentation task using the ADE20K dataset (Zhou et al., 2017b). Experiments on ADE20K use the decode head from UperNet (Xiao et al., 2018), and training recipes follow the default setting provided in mmsegmentation (Contributors, 2020). Here, the training resolution is set to $512 \times 512$, and the number of iterations is 160k. We evaluate models with different capacities in Table 7. We observe that our MUG approach results in significantly improved transfer performance across the ViT-S, ViT-B, and ViT-L backbones. We can see that the ViT-S MAE model only achieves 40.7% mIoU. Consistent, with the previous IN-1K linear probing results, MUG significantly alleviates the performance drop on the small backbone by improving mIoU by 2.6%. Notably when using the larger ViT-L backbone, MUG still improves over MAE by 2.1% mIoU. Compared to CLIP and MoCoV3, the superiority of MUG supports our information-theoretical motivation, *i.e.*, generative pre-training results in a better transferability upper bound compared to discriminative pre-training. Consequently, we observe that generative pre-training greatly benefits downstream fine-tuned performance.

## 5.3 Ablation Experiments

**Ablation: Impact of image reconstruction and text generation losses.** The most important ablation is the trade-off between the image reconstruction loss weight $\lambda_V$ and the caption generation loss $\lambda_L$. We fix $\lambda_V$ and ablate the value of $\lambda_L$. IN-1K fine-tuning experimental results are reported in Table 8. The best transfer performance is achieved when $\lambda_L = 0.1$, and both larger and smaller values perform worse. This illustrates the trade-off between image reconstruction and text generation.

Table 7: Semantic segmentation fine-tuned results (reported as mIoU) comparing different pre-training methods on ADE20K (Zhou et al., 2017b). MoCoV3* and MAE* are pre-trained on IN-1K.

| method | backbone | data | #epoch | mIoU |
|--------|----------|------|--------|------|
| MAE | ViT-S | CC3M | 400 | 40.7 |
| MUG (ours) | ViT-S | CC3M | 400 | **43.3** |
| CLIP | ViT-B | CC3M | 400 | 39.4 |
| SLIP | ViT-B | CC3M | 400 | 38.1 |
| MoCoV3* | ViT-B | IN-1K | 1600 | 47.3 |
| MAE* | ViT-B | IN-1K | 1600 | 48.1 |
| MoCoV3 | ViT-B | CC3M | 400 | 46.5 |
| MAE | ViT-B | CC3M | 400 | 45.5 |
| MUG (ours) | ViT-B | CC3M | 400 | **47.5** |
| MAE | ViT-L | CC3M | 400 | 50.0 |
| MUG (ours) | ViT-L | CC3M | 400 | **52.1** |

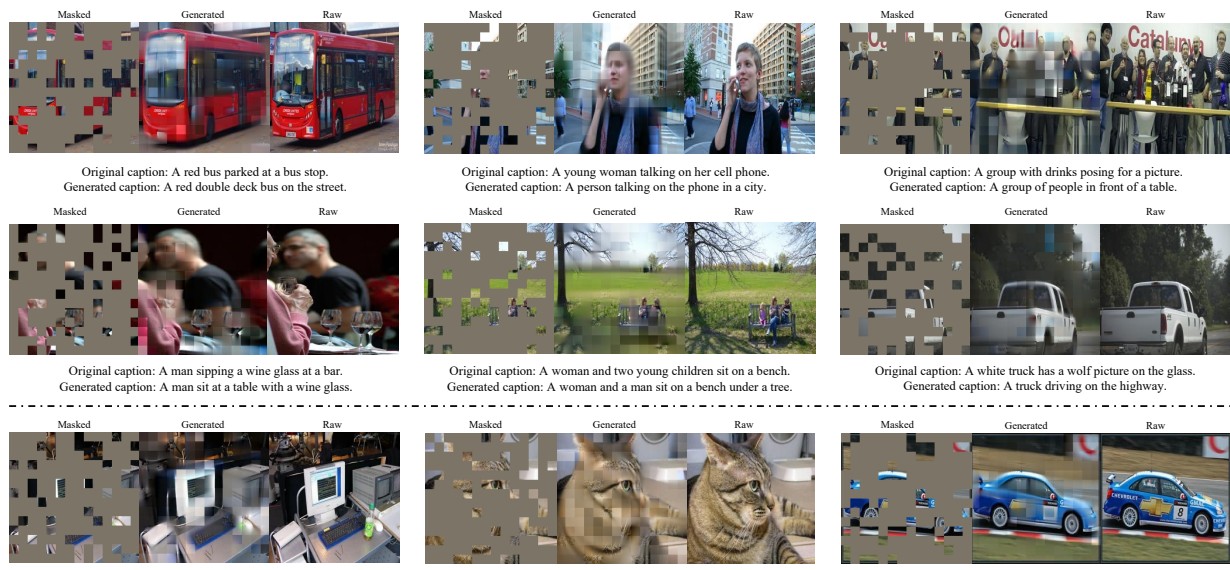

Figure 4: Reconstructions of masked images and captions from our MUG approach from the MS-COCO (top) and PASCAL-VOC (bottom) datasets.

Setting $\lambda_L = 0.0$ makes MUG equivalent to MAE. Increasing $\lambda_L$ in the range of (0.05, 0.3] increases the transfer performance, illustrating the effectiveness of our introduced text generation task. The reasons for the performance drop are two-fold: (i) As the CC3M pre-training data is web sourced, increasing $\lambda_L$ emphasizes the noisy text generation task. This can be problematic as captions coupled with web images can be often irrelevant, *e.g.*, an image contains a river, but its caption may be about hiking. (ii) As discussed in Section 3.2, the information contained in the text modality is greatly limited. Increasing $\lambda_L$ unavoidably leads to over-fitting, which is unfavorable in representation learning. Therefore, a trade-off between image reconstruction and text generation exists.

We further study the effects of the image reconstruction target. We simply set $\lambda_V$ to 0.0 and set $\lambda_L = 1.0$, which only permits the text generation task. Interestingly, setting $\lambda_V$ to 0.0 degrades MUG to our MAC baseline. Results are reported in Table 8 (right), where a notable performance drop is observed. This validates that (1) the information in text modality alone is insufficient, and (2) motivates the need for image generation for effective visual representation learning with image-text pairs.

**Ablation: Impact of the number of layers in text encoder.** For the text generation task, both single-modal and multi-modal layers are involved. We set the total number of layers to ten, and perform a grid search for the best result, see Table 9. We observe that setting the number of single-layers to one or two

Table 8: **Left:** Ablation of the language generation loss weight for MUG pre-trained on CC3M and fine-tuned on ImageNet-1k. **Right:** If we disable the visual generation task, performance suffers.

| $\lambda_L$ | 0.0 | 0.05 | 0.1 | 0.2 | 0.3 |
|---|---|---|---|---|---|
| ft. top-1 acc. | 83.0 | 83.3 | **83.5** | 83.2 | 83.0 |

| w/o $\mathcal{L}_V$ | w/ $\mathcal{L}_V$ |
|---|---|
| 81.7 | **83.5** |

achieves comparable performance. However, when the number of multi-layers decreases to five and six, the performance drop is significant. The reasons are likely two-fold: (i) A few single-modal layers can support the text generation task since the single-modal layers can serve as a key entity word extractor, which only has a small influence on visual representation learning. (ii) A certain number of multi-modal layers could be required to support the text generation task. An inadequate number of multi-modal layers further hurts the learning of image reconstruction because the performances becomes worse than the MAE baseline (83.0%).

Table 9: Ablation of the number of textual layers for MUG when pre-trained on CC3M and fine-tuned on ImageNet-1k.

| #single-/#multi-layers | 1/9 | 2/8 | 3/7 | 4/6 | 5/5 |
|---|---|---|---|---|---|
| F.T. top-1 acc. | 83.3 | **83.5** | 83.0 | 82.5 | 82.1 |

**Ablation: Impact of form of text generation.** Here we compare different text generation methods. There are two methods for text generation, *i.e.*, masked language modeling (MLM) and auto-regressive modeling (captioning). We set the optimal $\lambda_L$ values for both methods, and observe that captioning outperforms MLM by 0.5% top-1 accuracy (83.5% v.s. 83.0%) on IN-1k. A harder task could help to approach a higher $I(X^V, X^L; T)$. Captioning is regularized to sequentially predict words, while MLM is not regularized. The text generator is composed of single-modal and cross-modal parts, and the text generation methods actually affects the learning mechanism of the single-modal part. Concretely, the auto-regressive learning target forces the single-modal encoder to extract key words, *e.g.*, the word after an article could be a key entity noun. For masked language modeling, contextual words are possibly masked and predicted. However, such noisy and potentially meaningless reconstruction tasks likely do not benefit the vision encoder as much.

**Visualization: Reconstructed images with coupled text.** In Fig. 4, we provide generated images and captions produced by MUG on the MS-COCO (Lin et al., 2014) and PASCAL-VOC datasets (Everingham et al., 2015). The input to MUG only consists of a masked image and a start word (*e.g.*, "a"). As observed, MUG can recover the main objects and entities in the raw images and annotated captions, even if most of the image pixels are masked. It is reasonable that MUG only describes visible pixels, *e.g.*, in the middle of the second row, the left child is masked, and thus MUG only describes "a woman and a man". These visualizations show that the representation learned by MUG can generate the joint distribution $p(X^V, X^L)$.

# 6 Conclusion

In this work, we first benchmarked different self-supervised representation learning methods and outlined two main observations: (i) generative pre-training achieves the best transfer performance on web sourced datasets, and (ii) current multi-modal methods do not outperform single-modal ones. An information-theoretical view was developed to understand these benchmarking results and our observations. This inspired us to propose a new vision learner, titled MUlti-modal Generator (MUG), that makes use of web sourced image-text paired data. MUG learns visual latent representations using two generative objectives and demonstrates strong generalizability across multiple transfer learning tasks with satisfactory scaling performance, and validates our information-theoretical perspective.

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

# A Appendix

## A.1 Implementation Details

**Pre-training.** The default setting is in Table 10, and hyper-parameters mainly follow He et al. (2022) for fair comparisons. For each caption, we set the max length to 70 and use a percentage of 20% input words for processing. For each word, we mask it, replace it with a random word, or delete it with a probability of 50%, 10%, and 40%, respectively.

Table 10: Pre-training settings.

| config | value |
|---|---|
| optimizer | AdamW |
| base learning rate | 1.5e-4 |
| weight decay | 0.05 |
| optimizer momentum | $\beta_1, \beta_2 = 0.9, 0.95$ |
| batch size | 4096 |
| learning rate schedule | cosine decay |
| warmup epochs | 40 |
| image augmentation | RandomResizedCrop |
| text augmentation | mask/replace/delete |
| $\lambda_V$ | 1.0 |
| $\lambda_L$ | 0.1 |

**End-to-end fine-tuning on IN-1K, iNat-17, iNat-18, and P365.** We fine-tune models with the widely used recipe He et al. (2022); Bao et al. (2022) in Table 11 except for the layer-wise decay rate is set to 0.7. On all evaluated datasets and pre-training models, we use the same setting for fair comparisons.

**Linear probing.** Settings are provided in Table 12. We follow He et al. (2022) to introduce an extra BatchNorm layer without affine transformation for improving linear probing performances. For fair comparisons, all evaluated models are equipped with the extra layer.

**Semantic segmentation on ADE20K.** The ADE20K fine-tuning setting can be found in this website. We advise settings for ViT-S and ViT-L by following Touvron et al. (2020) and Bao et al. (2022), respectively. All evaluated pre-training models are fine-tuned with the same setting, and thus comparisons are fair.

Table 11: End-to-end fine-tuning settings.

| config | value |
|---|---|
| optimizer | AdamW |
| base learning rate | 1e-3 |
| weight decay | 0.05 |
| optimizer momentum | $\beta_1, \beta_2 = 0.9, 0.999$ |
| layer-wise lr decay | 0.7 |
| batch size | 1024 |
| learning rate schedule | cosine decay |
| warmup epochs | 5 |
| training epochs | 100 (B), 50 (L) |
| augmentation | RandAug (9, 0.5) |
| label smoothing | 0.1 |
| mixup | 0.8 |
| cutmix | 1.0 |
| drop path | 0.1 |

Table 12: Linear probing settings.

| config | value |
|---|---|
| optimizer | LARS |
| base learning rate | 0.1 |
| weight decay | 0 |
| optimizer momentum | 0.9 |
| batch size | 16384 |
| learning rate schedule | cosine decay |
| warmup epochs | 10 |
| training epochs | 90 |
| augmentation | RandomResizedCrop |

## A.2 Transfer Learning with Visual Prompts

Recent progresses in transfer learning in the visual domain has demonstrated the effectiveness of using visual prompt tuning (VPT) (Jia et al., 2022). By adding a small set of learnable tokens to the visual token sequence, VPT (Jia et al., 2022) can match, or even outperform, the performance of fully end-to-end fine-tuning of the pre-trained model. Thus VPT has been seen as an effective method for tuning a pre-trained model. In this section, we demonstrate that MUG learned representations are effective for use by VPT by evaluating on several fine-grained benchmarks from the original VPT paper (Jia et al., 2022). The results are presented in Table 13. We can see that MUG improves the performance on all the data that we tested, indicating that the features learned by MUG are not only effective for full fine-tuning, but also effective for parameter efficient tuning, which is more practical in real applications.

## A.3 More Diverse Evaluation

In this section, we provide linear probing results and visual prompt tuning results on additional datasets. These results complement the probing results in Table 4 and prompt tuning results in Table 13. We can

Table 13: Comparison classification performance using visual prompt tuning (Jia et al., 2022) across different pre-training methods and datasets.

| method | data | #epoch | CUB | SCars | NABirds |
|--------|------|--------|-----|-------|---------|
| MAE* | IN-1K | 1600 | 68.4 | 67.7 | 65.2 |
| MAE | CC3M | 400 | 69.7 | 68.8 | 67.1 |
| CLIP | CC3M | 400 | 67.8 | 68.4 | 66.3 |
| MUG (ours) | CC3M | 400 | **70.1** | 69.3 | 67.4 |
| MAE | W200M | 10 | 70.2 | 69.0 | 67.2 |
| CLIP | W200M | 10 | 69.0 | 68.7 | 66.1 |
| MUG (ours) | W200M | 10 | **70.5** | 69.5 | 68.0 |
| MAE | LAION400M | 5 | 70.4 | 69.6 | 68.1 |
| CLIP | LAION400M | 5 | 69.8 | 68.9 | 67.3 |
| MUG (ours) | LAION400M | 5 | **70.6** | 69.7 | 68.8 |

see that in terms of linear probing, both MAE and MUG do not achieve a good performance compared to other methods. However, in the visual prompt tuning experiments, MUG demonstrates strong performance. As demonstrated in the original MAE paper, linear probing results are not always correlated with the final transfer performance of the learned representations, as linear probing misses the opportunity to make use of strong, but non-linear, features. Thus, we are more interested in the visual prompt tuning results as they better demonstrate the learned representation's quality. The effective rank experiments in Table 16 also demonstrate that MUG can learn high quality representations.

Table 14: Linear probing results.

| Linear Probing | Data | Food101 | CIFAR100 | Aircraft | DTD | Pets | Caltech 101 |
|----------------|------|---------|----------|----------|-----|------|-------------|
| MoCoV3 | CC3M | 75.6 | 68.5 | 30.5 | 64.5 | 68.9 | 82.3 |
| CLIP | CC3M | 77.6 | 70.7 | 32.0 | 66.0 | 70.5 | 85.3 |
| SLIP | CC3M | 83.1 | 71.5 | 37.0 | 75.9 | 75.8 | 90.7 |
| MAE | CC3M | 70.4 | 63.4 | 25.6 | 61.3 | 63.1 | 76.9 |
| MUG | CC3M | 72.3 | 64.5 | 28.7 | 63.4 | 67.0 | 79.8 |

Table 15: Visual prompt tuning results.

| Visual Prompt Tuning | Data | Food101 | CIFAR100 | Aircraft | DTD | Pets | Caltech 101 |
|----------------------|------|---------|----------|----------|-----|------|-------------|
| MoCoV3 | CC3M | 81.4 | 76.7 | 37.6 | 71.2 | 84.2 | 86.1 |
| CLIP | CC3M | 83.5 | 77.5 | 38.9 | 74.4 | 85.6 | 88.3 |
| SLIP | CC3M | 85.6 | 77.9 | 39.4 | 76.1 | 86.3 | 90.4 |
| MAE | CC3M | 85.1 | 76.8 | 39.0 | 75.3 | 84.7 | 88.7 |
| MUG | CC3M | 86.7 | 78.4 | 41.3 | 77.4 | 85.3 | 91.9 |

## A.4 Assessing Transferability using RankMe

There have been a number of works that assess the performance of a pre-trained model on a downstream task prior to fine-tuning the model on the actual downstream dataset, *e.g.*, Garrido et al. (2022); Agrawal et al. (2022). RankMe (Garrido et al., 2022) is one such method that assesses the performance of a model using the effective rank of the features extracted by the model on a downstream task. This can be viewed as measuring part of the mutual information $I(X; Z)$, as the rank of the features describe the variation of the features across dimensions. $I(X; Z)$ describes not only the dimension-wise variation of the feature, but also the magnitude of variation within each of the dimensions.

As the measure of mutual information in high dimensional space is difficult (Belghazi et al., 2018), we leverage RankMe to show that MUG can indeed learn a superior visual representation. Here, this is measured in terms of having higher effective rank compared to other methods on features extracted from the ImageNet-1k dataset. The results are shown in Table 16. Overall, we observe that discriminative methods like MoCoV3

and CLIP have a lower effective rank than generative methods like MAE and MUG. Furthermore, MUG has the highest effective rank, indicating its ability to enlarge the information bottleneck of the model.

Table 16: Comparison of the effective rank of different pre-trained models on ImageNet-1k. Here, higher rank is better.

| method | data | #epoch | Rank |
|---|---|---|---|
| MoCoV3 | W200M | 10 | 413 |
| MAE | W200M | 10 | 674 |
| CLIP | W200M | 10 | 312 |
| MUG (ours) | W200M | 10 | **704** |
| MAE | LAION400M | 5 | 689 |
| CLIP | LAION400M | 5 | 425 |
| MUG (ours) | LAION400M | 5 | **792** |

## A.5  Pre-training on Larger Datasets

In this section we demonstrate the scability of MUG by pre-training models on larger datasets such as LAION400M (Schuhmann et al., 2021) and W200M. We mainly evaluate the performance of models using a ViT-B backbone. For pre-training models on different sized datasets, we keep the total number of iterations the same across different datasets for a fair comparison.

**Fine-tuning on ImageNet-1k.** We first compare the results of larger pre-training datasets for different pre-training methods on ImageNet-1k using end-to-end fine-tuning. Top-1 accuracy performance is reported in Table 17. We observe that similar trends from before still hold. Notably, MUG achieves the best results across IN-1k, IN-A, and IN-R. This demonstrate the scalability of MUG, as the performance of MUG still grows in line with the size of the pre-training dataset. In addition, MUG is significantly better than other multi-modal baselines.

Table 17: Comparison of methods on larger pre-training datasets on ImageNet-1k (IN-1K) using end-to-end fine-tuning.

| method | data | #epoch | IN-1K | IN-A | IN-R |
|---|---|---|---|---|---|
| MAE* | IN-1K | 1600 | 83.6 | 35.9 | 48.3 |
| MoCoV3 | W200M | 10 | 82.1 | 27.9 | 41.7 |
| MAE | W200M | 10 | 83.3 | 35.3 | 50.4 |
| CLIP | W200M | 10 | 79.8 | 25.6 | 42.7 |
| CoCa | W200M | 10 | 78.2 | 23.1 | 43.2 |
| MUG (ours) | W200M | 10 | **83.7** | **39.0** | **51.8** |
| MAE | LAION400M | 5 | 83.4 | 35.6 | 51.0 |
| CLIP | LAION400M | 5 | 80.1 | 29.7 | 45.2 |
| MUG (ours) | LAION400M | 5 | **83.9** | **42.1** | **54.7** |

Table 18: Comparison of different pre-training methods using larger datasets on ADE-20k using end-to-end semantic segmentation fine-tuning.

| method | data | #epoch | mIoU |
|---|---|---|---|
| MAE* | IN-1K | 1600 | 48.1 |
| MoCoV3 | W200M | 10 | 46.0 |
| MAE | W200M | 10 | 46.3 |
| CLIP | W200M | 10 | 40.3 |
| MUG (ours) | W200M | 10 | **48.5** |
| MAE | LAION400M | 5 | 48.9 |
| CLIP | LAION400M | 5 | 43.2 |
| MUG (ours) | LAION400M | 5 | **50.0** |

**Fine-tuning on ADE-20k.** We also provide a comparison using larger pre-training datasets when fine-tuning for semantic segmentation downstream task on ADE-20k. The results are presented in Table 18. It can be observed that MUG can indeed improve the transfer learning performance as the pre-training dataset grows. We note that MUG achieves the best transfer learning results, again indicating the effectiveness of MUG.

## A.6  Further Discussions on the Information Bottleneck View

In Section 3.2, we discussed about the inspirations of preliminary benchmark experiments. In this part, we extend the discussion from the following perspectives:

**Why does the single/multi-modal discriminative method have a narrow bottleneck?** Since popular discriminative methods are based on the contrastive learning, the nature of such methods could be

regarded as the few-shot classification task, and the reason is as follows. In each batch, we can regard each sample as a category, and the few-shot learning setting is B-way 1-shot, where B represents the batch size. For instance, in SimCLR (Chen et al., 2020b), an image is processed by two data augmentations as a positive pair, and "pulling close" the positive pair is actually minimizing a classification loss, *i.e.*, the cross entropy loss. We believe the information compression rate of a classification task is relatively large, since the model only requires the most representative patterns to distinguish defined categories. Information irrelevant to the classification task will be compressed in the training process. Therefore, regarding discriminative methods as classification tasks would explain the narrow information bottleneck, and thus the $I(X; Z_d)$ would be relatively small because of significantly compressed input information. In addition, it also explains that contrastive learning pre-training methods require a large batch size (Cui et al., 2022b), since increasing batch size will increase the difficulty of the few-shot classification task, which further alleviates the information compression. The narrow bottleneck also suggests that such methods require a certain amount of pre-training data, and insufficient data could result in the over-fitting problem.

**Why does the single-modal generative method have a wide bottleneck?** Popular single-modal generative methods are based on masked image modeling in an encoder-decoder fashion, and the working mechanism of such methods could be regarded as the image reconstruction task. The image reconstruction task is much more challenging than the contrastive learning (or few-shot classification) task, because the model is driven to preserve as much information as possible for recovering details (concrete pixel values) in the raw input image. It explains the wide bottleneck, since the decoder cannot finish the reconstruction task without sufficient information of the encoder's output. Thus, the $I(X; Z_g)$ would be larger than $I(X; Z_d)$, and the upper bound is $H(X)$. Therefore, better transferring results are achieved (He et al., 2022; Bao et al., 2022; Peng et al., 2022) by using single-modal generative methods than discriminative methods. The wide bottleneck also explains that generative methods can still perform well with a relatively small amount of pre-training data, as proven in He et al. (2022).

**Why does the multi-modal generative method have a wider bottleneck?** This work requires the model to reconstruct the raw input image and the coupled caption. It drives the model to preserve more information than the single-modal generative methods, since two objectives are involved and the decoder generates a joint distribution $p(X^V, X^L)$. The information bottleneck of a multi-modal generative method can be wider than that of a single-modal generative one. In this way, we drive the $I(X; Z)$ to approach $H(X^V, X^L)$, which has the potential to learn more transferable representations.

**How about scaling pre-training data?** Note that the above discussions are based on a hypothesis that the pre-training data is fixed, *e.g.*, CC3M in this work. Moreover, results in Section 5 prove that the transferring results of our proposed multi-modal generative method are the best. We further discuss the choice of pre-training methods when the pre-training data is scalable.

As aforementioned, single-modal discriminative methods could suffer from a large information compression rate; however, we can increase the amount of pre-training data as compensation. To prove this, we pre-train the SimCLR (Chen et al., 2020b) with our collected W200M data, and then we transfer the model via ImageNet-1k end-to-end fine-tuning. MAE (with W200M) and SimCLR (with W200M) achieve 83.3% and 83.4% top-1 accuracies, respectively. It means that, with large-scale data, the drawback of single-modal discriminative methods could be overcome. Besides, it also suggests that the scalability of single-modal generative methods can be inferior. However, our method MUG has good scalability by achieving 83.7%. Therefore, we suppose that our proposed method can perform well on different scales of pre-training data, which is an additional advantage.

### A.7   Additional Visualization Results

We further provide visualizations of the reconstructed images as well as the generated captions produced by our model on randomly selected images on the MS-COCO and PASCAL-VOC datasets in Figs. 5 and 6 for qualitatively understand the model's behaviour.

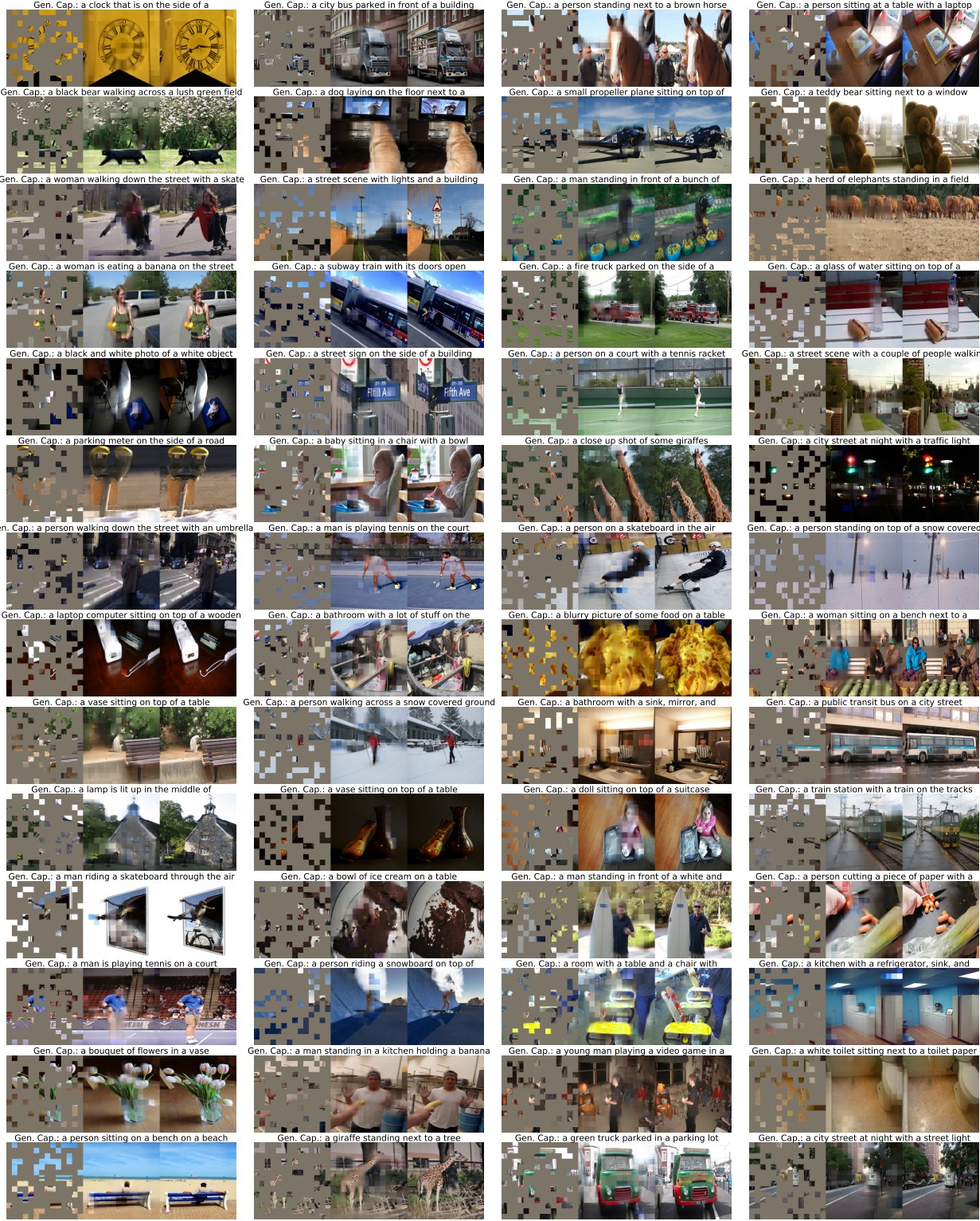

Figure 5: Uncurated random samples on COCO images. For each triplet, we show the masked image (left), MUG reconstruction (middle), the ground-truth (right), and the generated captions by MUG. The masking ratio is 75%.

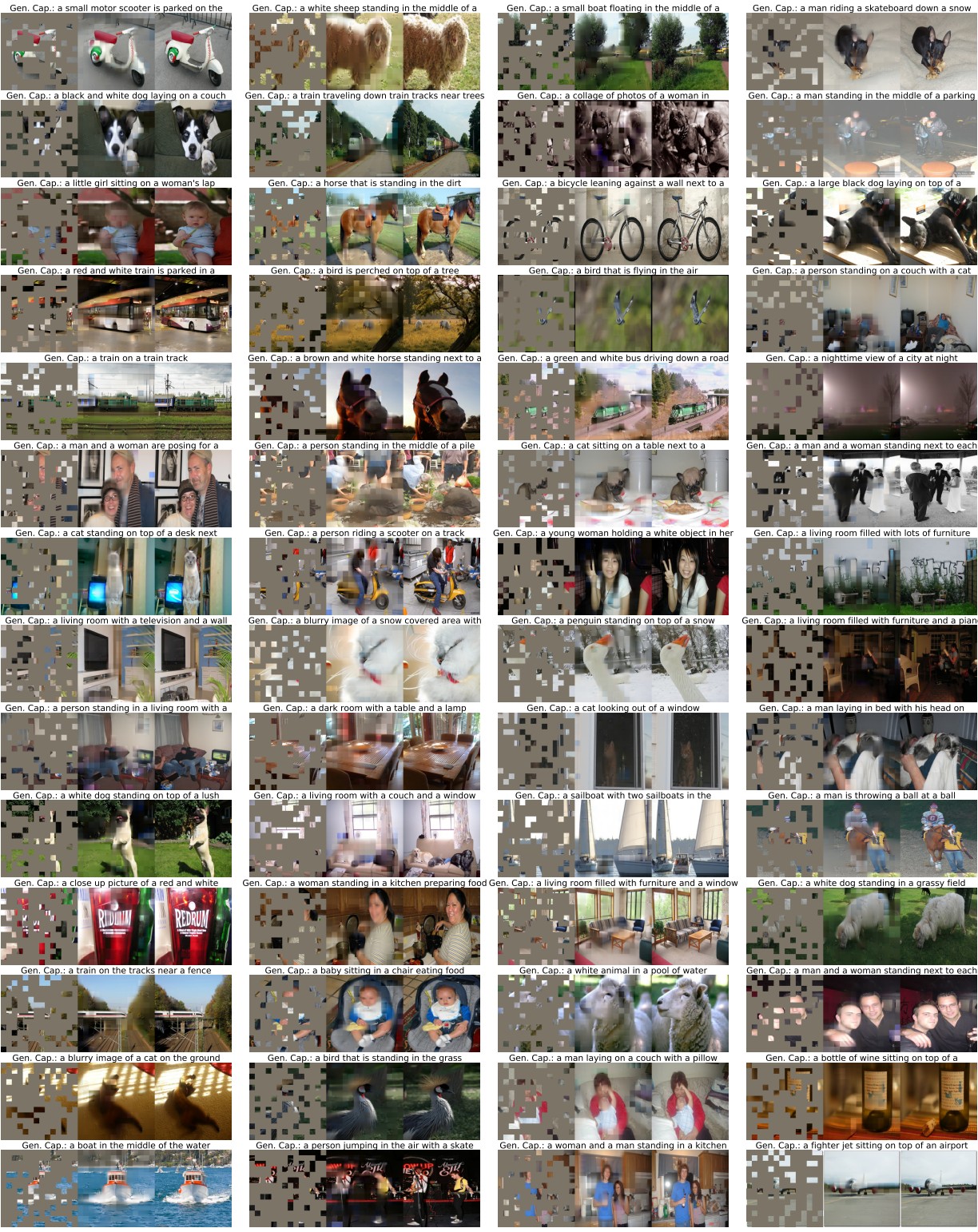

Figure 6: Uncurated random samples on PASCAL-VOC images. For each triplet, we show the masked image (left), MUG reconstruction (middle), the ground-truth (right), and the generated captions by MUG. The masking ratio is 75%.

### A.8 Limitations and Border Impact

One common limitation of models pre-trained on web sourced image-text pairs is that the models will inevitably be influenced by the open nature of internet data, such as biases or harmful speech. Thus the models may learn unwanted behaviors that may cause actual harm to people. Given this limitation, resulting models should be tested thoroughly before they are deployed in real-world applications.

