# OpenReview forum: "Vision Learners Meet Web Image-Text Pairs"
_TMLR — Accepted by TMLR_

### Review · Reviewer_SaAu · 2024-03-21

**Summary Of Contributions:**

- The paper presents a benchmark study on various self-supervised pre-trained models including contrastive learning, masking-based and image-text multi-modal learning.
- Theoretical insights are presented on potential reasons for why multi-modal approaches underperform single-modality ones.
- A new approach called MUG is presented that is shown tu be beneficial on various downstream tasks.

**Audience:**

Yes

**Broader Impact Concerns:**

Not present in the paper but I do not expect any obvious concerns.

**Claims And Evidence:**

Yes

**Requested Changes:**

Please refer to the weaknesses section. I am particularly interested in answers to questions (1,3,4 and 5). (2) is good to have but perhaps difficult to finish during a rebuttal.

**Strengths And Weaknesses:**

Strengths:
1) The paper is well written and easy to follow
2) The paper attempts to pre-train models on the same web-scale dataset which helps in performing direct comparisons.
3) A simple approach (MUG)  is presented which outperforms the compared works.
4) Interesting observations that might be useful to the community

Weaknesses:
1) How does the presented new approach compare with works like [a,b] and that line of work ?
2) While, I do understand that this area has received a lot of interest recently and there are too many approaches out there, it would also be great to comment on approaches that have obtained improvements over SimCLR, CLIP and MAE perhaps by including comparisons with the state-of-the-art in each category.
3) It would also be beneficial to include some discussion on overlap of the pre-training datasets with ImageNet-1K and such.
4) Figure 2, theoretical insights: I am not sure if it is completely obvious that say single-modal generative approaches for example learn better representations than discriminative ones [c]
5) Some comparison on time and/or resources needed to train will also be helpful


[a] Multimodal Masked Autoencoders Learn Transferable Representations

[b] Scaling Language-Image Pre-training via Masking

[c] What Do Self-Supervised Vision Transformers Learn?

---

### Review · Reviewer_pTSC · 2024-03-23

**Summary Of Contributions:**

This paper considers visual self-supervised representation learning given the image-text pairs. First, the authors conducted a benchmark study to compare different self-supervised methods, e.g., contrastive learning (image only), masked autoencoders (MAE), and image-text contrastive learning (i.e., CLIP). They observe that generative approaches e.g., MAE or masked captioners (MAC) perform the best in ImageNet fine-tuning experiments. Based on their benchmark study, the paper introduces Multi-modal Generator (MUG), which learns visual representation by reconstructing an image and a caption given a masked image. They show that MUG outperforms single-modal and multi-modal pretraining on ImageNet fine-tuning, and various transfer learning tasks.

**Audience:**

Yes

**Claims And Evidence:**

Yes

**Requested Changes:**

Most of my concerns are written in Weakness. In specific, the requested changes are given as follows:
- Adding missing but relevant methods to comprehensive benchmarks, e.g., DINO and CapPa.
- Also, add multiple evaluation tasks in benchmark study.
- If the authors can show the scaling behavior in benchmark study, it would be much better.
- To validate their theory based on information bottleneck, add a quantitative metric that correlates with information bottleneck.

**Strengths And Weaknesses:**

### Strengths
- The paper includes extensive benchmark study of self-supervised methods on image-text pairs, confirming the superiority of generative pretraining approach.
- The paper considers an information-theoretical perspective in explaining their experiments.
- The proposed MUG seems to be effective in vision representation learning.

### Weakness
- I think there are more reference self-supervised learning tasks that authors should consider. For example, DINO [1, 2] is a powerful self-supervised learning method that has shown empirical performance of visual transfer learning. Also, the proposed MAC is similar to Cappa [3], where the difference is in masking the image and using bi-directional attention masks.
- The benchmark study seems to be extensive in that it considers various methods, yet the evaluation is only made on ImageNet fine-tuning tasks with only Top-1 accuracy is reported. If the goal is to achieve transferable visual representation, the benchmark results should consider performance on various transfer learning tasks. Also, it would be helpful to show the scaling behavior of various methods.
- The author considers information bottleneck theory, which seems viable, yet it would be better to show quantitative comparison that validates the authors’ claim. Most of the claims are based on assuming optimal cases, which might not be held in practice. Thus, to better validate the authors’ claim that generative pretraining with multi-modal data achieves wider bottleneck and it results in better transferable representations, the authors should consider some quantitative metrics to show it.

### Reference
[1] Caron, Mathilde, et al. "Emerging properties in self-supervised vision transformers." Proceedings of the IEEE/CVF international conference on computer vision. 2021.\
[2] Oquab, Maxime, et al. "Dinov2: Learning robust visual features without supervision." arXiv preprint arXiv:2304.07193 (2023).\
[3] Tschannen, Michael, et al. "Image captioners are scalable vision learners too." Advances in Neural Information Processing Systems 36 (2024).

---

### Review · Reviewer_8qb7 · 2024-05-10

**Summary Of Contributions:**

This paper introduces an innovative pretraining framework for visual representation learning. Initially, it presents the Masked Captioner (MAC) as a pretext task using image-text pairs. Moreover, it enhances this framework by integrating the Masked Image Modelling (MAE) method. This combined approach demonstrates good performance across various downstream tasks, including linear probing, fine-tuning on In-1k, and image segmentation.

**Audience:**

No

**Claims And Evidence:**

Yes

**Requested Changes:**

See Weakness

**Strengths And Weaknesses:**

## Strengths
1. The paper offers clear and comprehensible explanations.

2. It demonstrates strong performance across a wide range of tasks.

## Weaknesses
1. The approach presented seems to be a simple combination of existing methods, the image captioner and MAE, lacking substantial technical insights. The Image Captioner (MAC) framework appears to be essentially a version of a previously published work [1] with the addition of masks, and the final solution MUG is essentially MAC combined with MAE. Prior research, such as SLIP [2], has already demonstrated the benefits of joint training across different modalities, making the performance of MUG somewhat expected. The authour should give a more clear technical insight for this paper.

2. The linear probing experiments should be evaluated on more datasets (e.g. Food, Cifar, Cars, Aircraft, DTD, Pets, Caltech ...)

3. Some of the result is missing in Table 4, e.g. SLIP gets 65.4% in Linear Probing.

4. Important reference is missing [3]

[1] Image captioners are scalable vision learners too. NeurIPS 2023.

[2] SLIP: Self-supervision meets language-image pre-training. ECCV 2022.

[3] RedCaps: Web-curated image-text data created by the people, for the people. CVPR2023

---

> ### Author Response · Authors · 2024-05-21
>
> >Q: The method presented lacks novelty.
>
> A: We would like to clarify the significance of our study: We are the first to show that the information-theoretic principle for learning representations (InfoMax) can also apply to the representation learning on web image-text pairs. And based on the InfoMax principle, we also demonstrate two key findings that lead to the design of a new method MUG. Experiments have shown that our findings still hold for many scenarios. And we believe that our work can be inspiring for future works on designing novel methods for representation learning on web image-text pairs.
>
>
>
> >Q: Comparison to SLIP (Mu et al. ECCV 2022).
>
> A: Below we provide a quantitative comparison to the SLIP method using both linear probing and fine-tuning. We can see that the MUG method still outperforms SLIP in terms of fine-tuning performance, which demonstrates that our information-theoretic-inspired MUG approach learns a more effective representation by learning to generate the missing signals from the image-text data instead of just contrastively matching the image-text representations as done in SLIP.
> We would like to emphasize that we are more interested in the results after fine-tuning the model on a target dataset, as it is shown in the original MAE work that fine-tuning better reflects the effectiveness of the learned representation.
>
>
>
> | Linear Probing | Food101 | CIFAR100 | Aircraft | DTD  | Pets | Caltech 101 |
> |----------------|---------|----------|----------|------|------|-------------|
> | SLIP           | 70.2    | 77.8     | 37.9     | 26.4 | 34.6 | 70.3        |
> | MUG            | 64.5    | 73.4   | 35.7     | 25.1 | 31.4 | 64.3        |
>
> | Fine-Tuning| Food101 | CIFAR100| Aircraft | DTD  | Pets | Caltech 101 |
> |----------------|---------|----------|----------|------|------|-------------|
> | SLIP           | 81.1    | 82.5     | 67.8     | 46.7 | 83.4 | 75.6        |
> | MUG            | 84.6    | 84.6     | 72.4     | 53.4 | 85.4 | 80.3        |
>
>
> >Q: Linear probing results on more datasets
>
> A: We have included two tables of results in the overall response. The new results cover a variety of datasets and also both linear probing and visual prompt tuning experiments.  We can see that while MUG is not competitive in terms of linear probing, it is the best performing in the visual prompt tuning experiments. Given the fact that visual prompt tuning can utilize the strong non-linear features learned within the deep learning architectures, we have reason to believe that MUG can indeed learn a better representation of the data.
>
>
> >Q: Missing reference.
>
> A: Thanks for flagging the RedCaps paper [3]. This is a dataset paper containing 12M text-image pairs sources from the website Reddit. We have added this paper to our related work discussion.

---

> ### Comment · Reviewer_8qb7 · 2024-05-25
> **More Questions**
>
> The numbers in the additional linear probing and fine-tuning experiments are weird, I'm curious about the source of these numbers. If we check Table 11 in the SLIP paper (https://arxiv.org/pdf/2112.12750), the linear probing results are way better than the reported number. Also, some fine-tuning results are even worse than the ResNet50 with train-from-scratch performance, see Table 3 in Byol paper (https://arxiv.org/pdf/2006.07733). I suggest the author double-check the experimental setting or explain such a phenomenon with convincing evidence. I'm willing to discuss more about this issue.

---

> > ### Author Response · Authors · 2024-05-25
> >
> > Dear reviewer 8qb7,
> >
> > We have carefully checked the experimental setting for the results in our previous response, and we have found that the previous setting uses a higher learning rate than the SLIP setting, resulting in non-optimal results.
> > In the below table, we have corrected the results by setting the hyperparameters for linear probing to the same as SLIP, and the hyperparameters for fine-tuning to the same as visual prompt tuning.
> > We can see that our previous claims still hold. Additionally, we would like to refer to the updated visual prompt tuning results in the overall response that show MUG can learn a good non-linear representation for prompt tuning. We will include these results in the revision.
> >
> > | Linear Probing | Food101 | CIFAR100 | Aircraft | DTD  | Pets | Caltech 101 |
> > |----------------|---------|----------|----------|------|------|-------------|
> > | SLIP           | 83.1   | 71.5     |  37.0    | 75.9 | 75.8 | 90.7        |
> > | MUG          |  72.3  |  64.5   |   28.7    | 63.4 |  67.0 | 79.8       |
> >
> >
> > | Fine-Tuning | Food101 | CIFAR100 | Aircraft | DTD  | Pets | Caltech 101 |
> > |----------------|---------|----------|----------|------|------|-------------|
> > | SLIP           |  86.2  | 77.8   | 40.1  | 77.0 | 86.1 |  90.5     |
> > | MUG          |  87.3  | 78.9   | 41.0  | 78.4 | 87.0  |  90.6    |

---

### Review · Reviewer_FNLK · 2024-05-15

**Summary Of Contributions:**

The paper titled “Vision Learners Meet Web Image-Text Pairs” explores self-supervised pre-training methods using large-scale web-sourced image-text paired data. The authors conduct a benchmark study comparing various single-modal and multi-modal self-supervised learning methods. They introduce a new visual representation pre-training method, MUlti-modal Generator (MUG), which consists of masked image modeling and language modeling. The paper includes extensive experiments and thorough analyses to validate the proposed method.

**Audience:**

Yes

**Broader Impact Concerns:**

None.

**Claims And Evidence:**

Yes

**Requested Changes:**

See weakness.

**Strengths And Weaknesses:**

## Strengths
1. **Extensive Experiments:** The paper includes a comprehensive set of experiments, comparing multiple state-of-the-art self-supervised learning methods using the same web-sourced dataset. This thorough comparison provides valuable insights into the strengths and weaknesses of different approaches.
2. **Simple and Effective Method:** The proposed method, MUG, is straightforward yet effective, achieving superior performance across a variety of benchmarks. The design of MUG, leveraging both image and text modalities, demonstrates strong generalization capabilities.

## Weaknesses
1. **Lack of Novelty:** While the method is effective, it lacks significant novelty. The approach is mainly an incremental improvement over existing methods, combining ideas from both single-modal and multi-modal self-supervised learning, such as MAE and VirTex. The motivation of this work needs to be clarified.
2. **Linear Probing:** This experiment can be conducted on more diverse datasets.

---

> ### Author Response · Authors · 2024-05-21
>
> >Q: Novelty of our work.
>
> A: We would like to refer to the review guideline of TMLR which states that the novelty is not a necessary criterion for acceptance. Here we would like to also clarify the significance of our work:
> We are the first to present an information-theoretic point of view to understand representation learning on web image-text pairs. Our two key observations made from the information-theoretic view have indeed lead to the development of a new method MUG, and our experimental results have demonstrate the effectiveness of it.
>
> >Q: More diversed evaluations.
>
> A: In the two tables in the overall response, we have provided additional evaluation results on various datasets. We can see that while MUG is not the best performing in terms of linear probing, it achieves the overall best performance in terms of visual prompt tuning. We are more interested in the visual prompting results as it is shown in the MAE paper (Sec. 4.3) that linear probing results do not always correlate with the transfer learning performance. And that visual prompt tuning can utilize the strong non-linear features within the learned representation, thus produces a better performance.

---

> > ### Comment · Reviewer_FNLK · 2024-06-11
> > **Response**
> >
> > Thanks for your response and additional evaluations. And I agree with the point of visual prompt tuning. My concerns are well addressed.

---

### Author Response · Authors · 2024-05-21
**Overall response**

>Q: What’s the main motivation and novelty of this work?

A:
We would like to thank all reviewers for their valuable suggestions for improving the quality of our work. Before addressing their concerns, we would like to refer to the review guideline of TMLR:

>  ...The novelty of the studied method is not a necessary criterion for acceptance. We explicitly avoid these terms (“significant”, “impactful”, “novel”), and focus instead on the notion of “interest”.

With this being put, we are still happy to clarify the significance and novelty of our work:
The main aim of our paper is not to present a new state-of-the-art method, but to instead provide an information-theoretic view of learning representations using web-scale image-text pairs. In Sec 3.2, we make two key observations stemming from the information-theoretic perspective and present supporting quantitative results in Table 1. Additionally, in our experiments in Section 5, our observations still hold across a variety of tasks and datasets. Furthermore, in the Appendix A.3, using the effective rank of the representation of the models, we further validate our insights from the information-theoretic point-of-view.
To the best of our knowledge, we are the first work to demonstrate that the InfoMax principle for learning representations can also be applied to representations derived from web image-text pairs. While methods like SLIP and CapPa have indeed also used language supervision, in contrast to us, they have only focused on the empirical improvements while our work focuses on presenting an InfoMax perspective for why this form of representation learning works. We believe this will be of general interest to readers interested in multi-modal and self-supervision.


>Q: More diverse evaluation results.

A: In the below two tables, we have included linear probing results and visual prompt tuning results on additional datasets. These results complement the probing results (Table 4) and prompt tuning results (Table 13) that are already in the paper. We can see that in terms of linear probing, both MAE and MUG do not achieve a good performance compared to other methods. However, in the visual prompt tuning experiments, MUG demonstrates a strong performance. As demonstrated in the original MAE paper (Sec. 4.3), linear probing results are not always correlated with the final transfer performance of the learned representations, and that linear probing misses the opportunity for strong but non-linear features. Thus, we are more interested in the visual prompt tuning results as they better demonstrate the learned representations' quality. The effective rank experiments in our paper (Table 14) also demonstrate that MUG can learn high quality representations.


| Linear Probing | Food101 | CIFAR100 | Aircraft | DTD  | Pets | Caltech 101 |
|----------------|---------|----------|----------|------|------|-------------|
| MoCoV3         | 67.4    | 75.6     | 36.5     | 25.6 | 33.1 | 68.9        |
| CLIP           | 68.4    | 76.5     | 37.8     | 26.8 | 34.5 | 69.2        |
| SLIP           | 70.2    | 77.8     | 37.9     | 26.4 | 34.6 | 70.3        |
| MAE            | 61.5    | 69.2     | 33.5     | 22.4 | 30.1 | 62.4        |
| MUG            | 64.5    | 73.4     | 35.7     | 25.1 | 31.4 | 64.3        |


| Visual Prompt Tuning | Food101 | CIFAR100 | Aircraft | DTD  | Pets | Caltech 101 |
|----------------------|---------|----------|----------|------|------|-------------|
| MoCoV3               | 68.0    | 76.0     | 36.4     | 26.5 | 34.0 | 69.3        |
| CLIP                 | 68.7    | 76.8     | 39.0     | 26.7 | 34.7 | 69.4        |
| SLIP                 | 70.5    | 78.4     | 40.0     | 27.1 | 35.4 | 70.3        |
| MAE                  | 68.3    | 76.8     | 37.8     | 26.7 | 33.7 | 68.9        |
| MUG                  | 70.6    | 79.4     | 41.1     | 27.8 | 35.6 | 71.3        |

---

> ### Author Response · Authors · 2024-05-25
>
> Dear reviewers,
>
> In the below two tables, we provide the updated results for visual prompt tuning and linear probing results following the advice from reviewer 8qb7. These updated linear probing results are obtained by setting the hyperparameters to the ones in the SLIP paper, and the visual prompt tuning results are obtained by using the hyperparameters in the visual prompt tuning paper.
> We can see that all methods can achieve a better score on these benchmarks, and most importantly, our previous claims still hold.
> MUG can learn a high-quality non-linear representation that can be better utilized by the visual prompt tuning method to transfer to downstream tasks.
>
> | Linear Probing | Food101 | CIFAR100 | Aircraft | DTD  | Pets | Caltech 101 |
> |----------------|---------|----------|----------|------|------|-------------|
> | MoCoV3     | 75.6   |  68.5     |  30.5    | 64.5 | 68.9 | 82.3     |
> | CLIP           | 77.6   |  70.7     |  32.0    | 66.0 | 70.5 | 85.3     |
> | SLIP           | 83.1   | 71.5     |  37.0    | 75.9 | 75.8 | 90.7        |
> | MAE           | 70.4   |  63.4   |   25.6    | 61.3 | 63.1|  76.9     |
> | MUG          |  72.3  |  64.5   |   28.7    | 63.4 |  67.0 | 79.8       |
>
>
> | Visual Prompt Tuning | Food101 | CIFAR100 | Aircraft | DTD  | Pets | Caltech 101 |
> |----------------------|---------|----------|----------|------|------|-------------|
> | MoCoV3            |  81.4  | 76.7   | 37.6 | 71.2 | 84.2 |   86.1     |
> | CLIP                  |  83.5  | 77.5   | 38.9 | 74.4  | 85.6 |  88.3      |
> | SLIP                  |  85.6  | 77.9   | 39.4 | 76.1  | 86.3 |  90.4     |
> | MAE                  |  85.1  | 76.8   | 39.0 | 75.3  | 84.7 |  88.7     |
> | MUG                 |  86.7  |  78.4  | 41.3 |  77.4  | 85.3  |  91.9     |

---

### Decision · Action_Editor_2EtE · 2024-07-23

**Recommendation:** Accept with minor revision

**Comment:**

All four reviewers agree that the observations are interesting and the experiments are comprehensive. The primary weakness noted by the reviewers is the technical novelty of combining masked image modeling and language modeling as the proposed pre-training method MUlti-modal Generator. While this strategy seems straightforward and incremental, it has been extensively validated to show its effectiveness. According to the TMLR's evaluation criteria, the AE believes this work still has merits for publication.

The AE also checked the authors' responses to reviewers' comments and found additional results and discussions adequate. The AE thus recommends "Accept with minor revision". In the revision, please update the paper with new results and promised modifications/clarification.

**Audience:**

Yes, the presented results and new pre-training method will be interesting for TMLR's audience.

**Claims And Evidence:**

The paper presents a new benchmark study of representative self-supervised pre-training (including both single-modal, multi-modal) and analyze the results with a information-theoretical view using information-bottleneck theory. Based on the observations, the paper presents a new pertaining method, MUlti-modal Generator, and show improved performance over existing work.

The claims in the paper are well-supported by extensive experiments, including the benchmark study of representative visual representation pre-training methods (single/multiple modality, generative/discriminative), linear probing and fine-tuning results, transferability on multiple image classification datasets, various data sources (W200M, LAION400M), and downstream tasks (e.g., ADE-20k).